# A tutorial on how not to over-interpret STRUCTURE and ADMIXTURE bar plots

Daniel J. Lawson [1], Lucy van Dorp[2,3] & Daniel Falush[4]

Genetic clustering algorithms, implemented in programs such as STRUCTURE and ADMIX-TURE, have been used extensively in the characterisation of individuals and populations based on genetic data. A successful example is the reconstruction of the genetic history of African Americans as a product of recent admixture between highly differentiated populations. Histories can also be reconstructed using the same procedure for groups that do not have admixture in their recent history, where recent genetic drift is strong or that deviate in other ways from the underlying inference model. Unfortunately, such histories can be misleading. We have implemented an approach, badMIXTURE, to assess the goodness of fit of the model using the ancestry "palettes" estimated by CHROMOPAINTER and apply it to both simulated data and real case studies. Combining these complementary analyses with additional methods that are designed to test specific hypotheses allows a richer and more robust analysis of recent demographic history.

[1] University of Bristol, Integrative Epidemiology Unit, Population Health Sciences, Bristol BS8 1TH, UK. [2] University College London Genetics Institute (UGI), University College London, London WC1E 6BT, UK. [3] Centre for Mathematics and Physics in the Life Sciences and Experimental Biology (CoMPLEX), University College London, London WC1E 6BT, UK. [4] Milner Centre for Evolution, University of Bath, Bath BA2 7AY, UK. Correspondence and requests for materials should be addressed to D.F. (email: danielfalush@googlemail.com)

Model-based clustering has become a popular approach to visualise the genetic ancestry of humans and other organisms. Pritchard et al.[1] introduced a Bayesian algorithm STRUCTURE for defining populations and assigning individuals to them. FRAPPE and ADMIXTURE were later implemented based on a similar underlying inference model but with algorithmic refinements that allow them to be run on data sets with hundreds of thousands of genetic markers[2,3]. Following many successful examples of inference[4–6], the STRUCTURE barplot has become a de-facto standard used as a non-parametric description of genetic data[7] alongside a Principle Components Analysis[8]. However, some experienced researchers feel that STRUCTURE has become "a victim of its own success" due to frequent over-interpretation of the results[7].

Experienced researchers, particularly those interested in population structure and historical inference, typically present STRUCTURE results alongside other methods that make different modelling assumptions. These include TreeMix[9], ADMIXTUREGRAPH[10], fineSTRUCTURE[11], GLOBETROTTER[12], f3 and D statistics[13], amongst many others. These models can be used both to probe whether assumptions of the model are likely to hold and to validate specific features of the results. Each also comes with its own pitfalls and difficulties of interpretation. It is not obvious that any single approach represents a direct replacement as a data summary tool. Here we build more directly on the results of STRUCTURE/ADMIXTURE by developing a new approach, badMIXTURE, to examine which features of the data are poorly fit by the model. Rather than intending to replace more specific or sophisticated analyses, we hope to encourage their use by making the limitations of the initial analysis clearer.

## Results

**The default interpretation protocol.** Most researchers are cautious but literal in their interpretation of STRUCTURE and ADMIXTURE results, as caricatured in Fig. 1, as it is difficult to interpret the results at all without making several of these assumptions. Here we use simulated and real data to illustrate how following this protocol can lead to inference of false histories, and how badMIXTURE can be used to examine model fit and avoid common pitfalls.

**Case study 1: African Americans.** In order to understand the assumptions underlying the STRUCTURE model, it is helpful to think about an example that was originally used to motivate it, namely the ancestry and genetic history of African Americans. The "admixture model" of STRUCTURE assumes that each individual has ancestry from one or more of $K$ genetically distinct sources. In the case of African Americans, the most important sources are West Africans, who were brought to the Americas as slaves, and European settlers. The two groups are thought to have been previously separated with minimal genetic contact for tens of thousands of years. This means that their history can be separated into two phases, a "divergence phase" lasting thousands of years of largely independent evolution and an "admixture phase", in which large populations met and admixed within the last few hundred years. Specifically, most of the ancestors of African Americans that lived 500 years ago were either Africans or Europeans. The goal of the algorithm is to reconstruct the gene frequencies of these two distinct "ancestral" populations and to estimate what proportion of their genome each African American inherited from them.

When the STRUCTURE admixture model is applied to a data set consisting of genetic markers from West Africans, African Americans and Europeans it infers two ancestral populations[1]. Each of the Europeans and Africans are assigned a great majority of their ancestry from one of them. African–Americans are inferred to have an average of 18% ancestry from the European cluster but with substantial inter-individual variation[14].

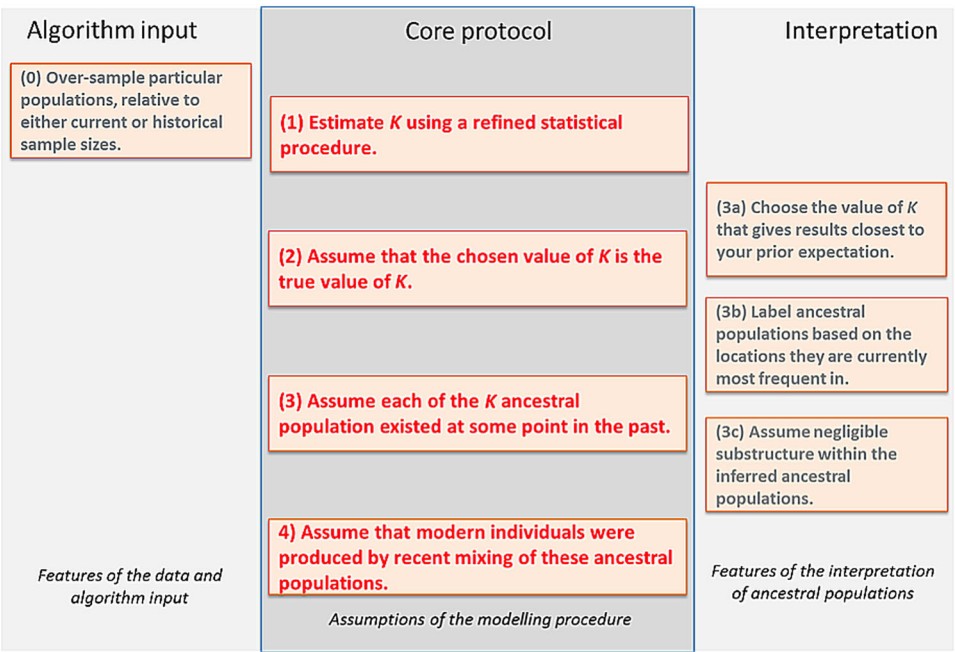

**Fig. 1** A protocol for interpreting admixture estimates, based on the assumption that the model underlying the inference is correct. If these assumptions are not validated, there is substantial danger of over-interpretation. The "Core protocol" describes the assumptions that are made by the admixture model itself (Protocol 1, 3, 4), and inference for estimating $K$ (Protocol 2). The "Algorithm input" protocol describes choices that can further bias results, while the "Interpretation" protocol describes assumptions that can be made in interpreting the output that are not directly supported by model inference

Assignment of clusters in this case is readily biologically interpretable. There are of course genetic differences amongst both the Africans and the Europeans who contributed to African–American ancestry, e.g., reflecting genetic variation between regions within Europe and Africa, but the divergence between Europeans and Africans took place over millennia and is of a different magnitude. These subtle differences are likely to have a relatively minor effect on the amount of African and European ancestry estimated for each individual. Therefore the STRUCTURE admixture proportion is a reasonably accurate estimate of the recent admixture fraction.

**Different scenarios give indistinguishable ADMIXTURE plots.** Many real population histories are not neatly separable into divergence and admixture phases but the methods can be applied to any data set, producing ancestry bar plots. Figure 2 shows admixture histories inferred by ADMIXTURE for three demographic scenarios. These simulations were performed with 13 populations (see "Methods" section)—which provides valuable out-group information—but only results for the four most relevant populations are shown. The "Recent Admixture" scenario represents a history qualitatively similar to African Americans, in which the admixture model holds. The true history is that P2 is an admixture of P1, P3 and P4. ADMIXTURE, interpreted according to the protocol, infers that this is what happened and estimates approximately correct admixture proportions (true admixture proportions are 35% light green and 15% light pink).

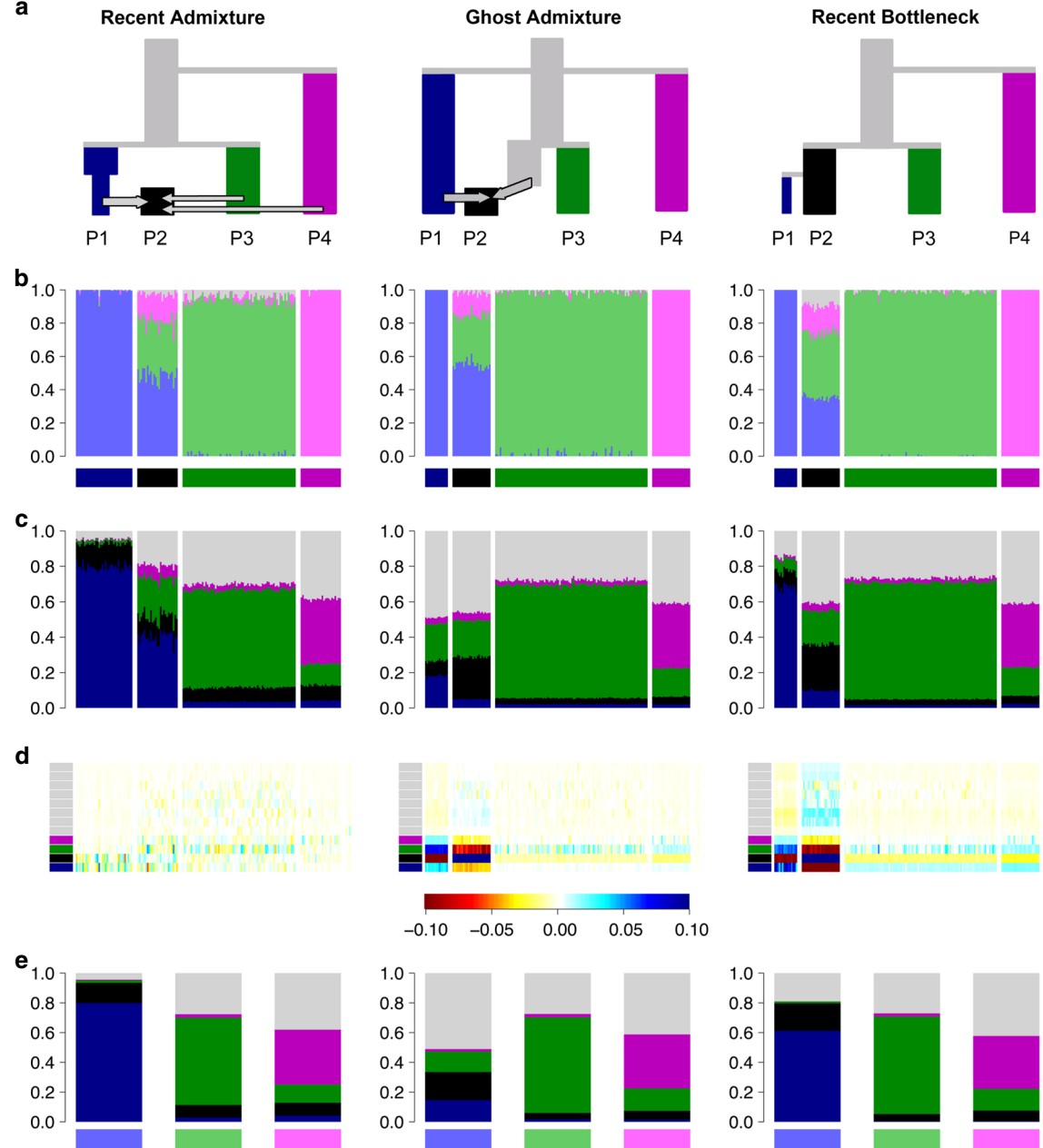

**Fig. 2** Three scenarios that give indistinguishable ADMIXTURE results. **a** Simplified schematic of each simulation scenario. **b** Inferred ADMIXTURE plots at $K = 11$. **c** CHROMOPAINTER inferred painting palettes. **d** Painting residuals after fitting optimal ancestral palettes using badMIXTURE, on the residual scale shown. **e** Ancestral palettes estimated by badMIXTURE. 13 populations in total were simulated, with grey populations all being outgroups to those shown in colour

In the "Ghost Admixture" scenario, P2 is instead formed by a 50–50% admixture between P1 and an unsampled "ghost" population, which is most closely related to P3. In the "Recent Bottleneck" scenario, P1 is a sister population to P2 that underwent a strong recent bottleneck. The ADMIXTURE plots are almost identical between the three scenarios. Since the "Ghost Admixture" and "Recent Bottleneck" scenarios cannot be represented using a simple admixture description, the model cannot be historically correct. Nevertheless, the algorithm attempts to fit the data as best it can by finding the combination of admixture proportions and ancestral frequencies that best explain the observed patterns.

To be specific, in the Ghost Admixture scenario of Fig. 2, the ghost population is modelled as a mix of the sampled populations it is most closely related to, rather than being given its own ancestral population. For this scenario, the larger proportion of ancestry inferred from the light green population rather than the light pink one does not reflect a difference in admixture proportions, since neither P3 nor P4 actually contributed genetic material to P2. Rather, it reflects the fact that P3 is more closely related to the unsampled ghost population, as seen in the phylogeny.

In the Recent Bottleneck scenario, ADMIXTURE models the genetic drift shared by P1 and P2 by assigning both populations some ancestry from the light blue ancestral population. The strong recent drift specific to P1 is approximately modelled by assigning more light blue ancestry to P1 than to P2, thereby making P1 more distinct from the other populations in the sample. The remaining ancestries assigned to P2 are from the most closely related of the remaining ancestry components, again in proportions that reflect phylogenetic distance rather than admixture fractions. An alternative outcome in both scenarios would be for ADMIXTURE to infer a higher value of $K$ and to include an extra ancestral population for P2. The algorithm is more likely to infer this solution if there was stronger genetic drift specific to P2 or if members of the population made up a greater overall proportion of the sample.

**badMIXTURE results distinguish between scenarios.** badMIXTURE uses patterns of DNA sharing to assess the goodness of fit of a recent admixture model to the underlying the genetic data. These sharing profiles are generated using CHROMO-PAINTER[11], which calculates, for each individual, which of the other individual(s) in the sample are most closely related for each stretch of genome, using either haplotype or allele matching. This process is called "chromosome painting", and can be thought of in terms of "palettes" (Fig. 2c), which can also be visualised as bar plots. The palette measures the proportion of the genome of each individual that is most closely related to the individuals sampled from each of the labelled populations. The painting palettes differ for the three simulated scenarios (Fig. 2c), showing that there should be information in the genetic data to distinguish between them, even though they give almost identical ADMIXTURE bar plots.

STRUCTURE and ADMIXTURE estimate both the ancestral gene frequencies and the admixture proportions for each individual in the sample. badMIXTURE assumes that the admixture proportions estimated by STRUCTURE and ADMIXTURE are correct and uses matrix factorisation to find the combination of ancestral palettes that give the best overall fit (evaluated using least squares) to the palettes of each individual. Crucially, under a number of reasonable assumptions (see Methods), in a recent admixture scenario, the palettes of admixed individuals should be a mixture of the palettes of non-admixed individuals according to the relevant admixture proportions.

In other words, if a simple admixture scenario is correct and the proportions are correctly estimated by STRUCTURE/ADMIXTURE, then it should be possible to use the $N \times K$ admixture proportions of the $N$ individuals in the sample and the $K \times P$ palettes proportions for the $K$ ancestral populations to predict the $N \times P$ palette proportions for each individual. The fit of the model can be examined by comparing the true palette proportions for each individuals to the ones predicted by badMIXTURE. An admixture model can only be a parsimonious way of describing the data if there are more distinct ancestry profiles than there are ancestral populations, since otherwise each ancestry profile could simply be assigned its own ancestral population. Therefore, badMIXTURE assumes there are more distinct ancestry profiles $P$ than there are populations $K$.

Figure 2d shows the residuals, representing the difference between the observed palettes for each individual in the simulated data and those reconstructed by badMIXTURE. Figure 2e shows the corresponding palettes inferred for each ancestral population. Under the Recent Admixture scenario, there is no systematic pattern to the residuals. For the Ghost Admixture scenario, the residuals show a systematic pattern, with the model substantially underestimating the proportion of palette that individuals in P2 have from their own population and overestimating the contributions from the other populations. For the Recent Bottleneck model, the deviations are similar—the main qualitative difference between the Ghost Admixture scenario and Recent Bottleneck scenario are in the ancestral palettes. Ghost Admixture produces a more uniform ancestral palette than either of the other models, which both contain bottlenecks for P1.

badMIXTURE distinguishes the Recent Admixture scenario from alternatives because the Recent Admixture model makes the distinct prediction that admixed individuals are not particularly related to each other, as shown by the small amount of black in their palettes in Fig. 2c. Members of P2 get 50% of their genomes from the light blue ancestral population, 35% from the light green population and 15% from the light pink one, while P1 received all of its ancestry from the light blue population. For any given locus, a member of P2 will have the same ancestral source as a member of P1 50% of the time. However, two members of P2 will have the same ancestry source only $0.5^2 + 0.35^2 + 0.15^2 = 0.395$ of the time. This means that paradoxically, members of P2 may (depending on the exact details of population history) be more related to members of P1 than they are to each other and have relatively little of their palette from their own population. Under the other scenarios, individuals from P2 receive more of their palette from other members of their own population.

**badMIXTURE is still informative without linkage information.** STRUCTURE/ADMIXTURE has been applied to thousands of different species, most of which do not have the linkage maps (either physical or genetic) usually required for chromosome painting. The algorithm can also be applied to data sets with relatively small numbers of markers. It would therefore be advantageous to be able to apply a similar approach to these data sets.

To evaluate fit using such types of data, chromosome painting can be implemented using an unlinked model, as shown in Fig. 3, to generate allele-sharing palettes. The results are qualitatively similar to the CHROMOPAINTER analysis exploiting Linkage Disequilibrium; however, because the palettes are closer to uniform (Fig. 3b), the residuals contain more noise (Fig. 3c). If few markers were available, there may be no interpretable signal remaining making it impossible to distinguish between different scenarios on the basis of limited genetic information.

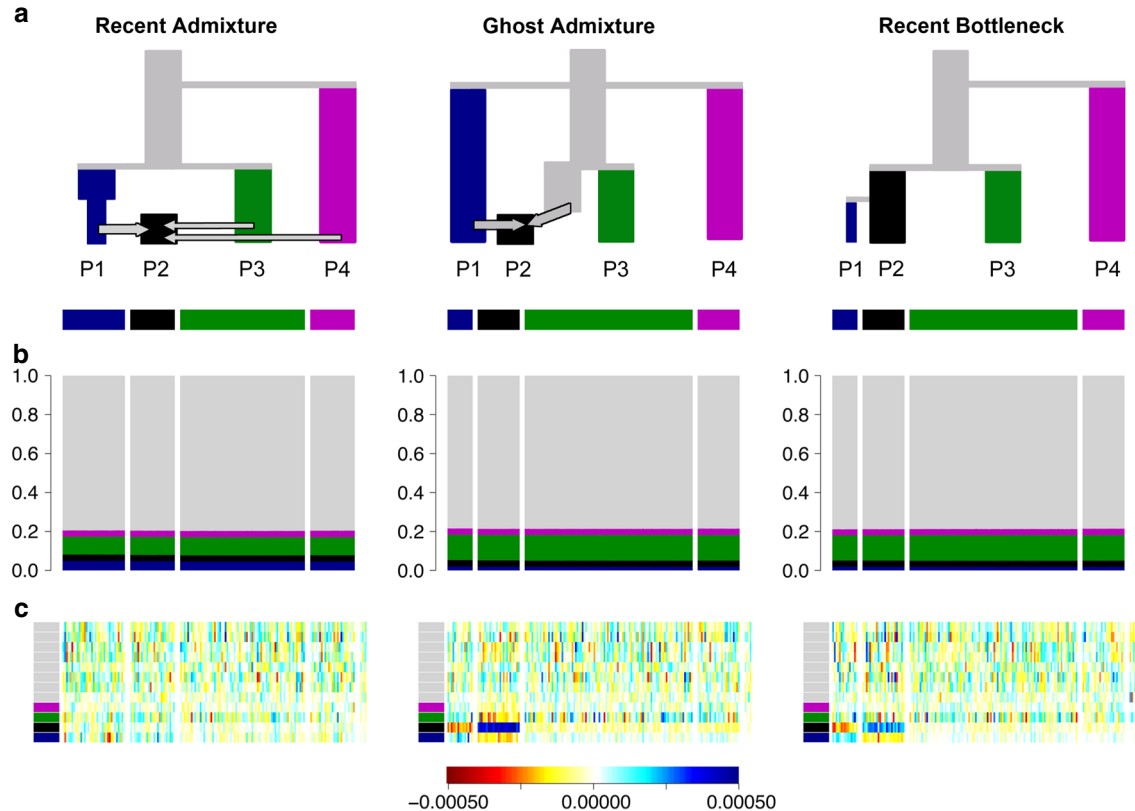

**Fig. 3** Unlinked badMIXTURE results for simulated data. Same scenario (**a**) and data as presented in Fig. 2 but assuming markers are unlinked. While the palettes look dramatically more homogeneous without linkage information (**b** vs. Fig. 2c), the badMIXTURE residuals (**c** vs Fig. 2d) follow the same pattern, i.e., they are unstructured in the Recent Admixture data (scale shown below main plots)

**Case study 2: The Ari of Ethiopia**. This case study highlights a situation in which application of badMIXTURE could have prevented a false history from being inferred. Three sets of researchers[15–17] investigated the relationships between the origins of occupational groups (Blacksmiths and Cultivators) in the Ari community of Ethiopia, all applying ADMIXTURE analyses (Fig. 4a–c). The first two sets of researchers tentatively concluded that the two groups were most likely to derive from different ancestral sources.

First, Pagani et al.[15] analysed the data and cautiously interpreted the ADMIXTURE results:

"One insight provided by the ADMIXTURE plot (Fig. 4a) concerns the origin of the Ari Blacksmiths. This population is one of the occupational caste-like groups present in many Ethiopian societies that have traditionally been explained as either remnants of hunter-gatherer groups assimilated by the expansion of farmers in the Neolithic period or as groups marginalised in agriculturalist communities due to their craft skills. The prevalence of an Ethiopian-specific cluster (yellow in Fig. 4a) in the Ari Blacksmith sample could favour the former scenario; the ancestors of this occupational group could have been part of a population that inhabited the area before the spread of agriculturalists."

This interpretation was supported by a similar analysis by Hodgson et al.[17]:

"As the Ari Blacksmiths have negligible EthioSomali ancestry, it seems most likely that the Ari Cultivators are the descendants of a more recent admixture between a population like the Ari Blacksmiths and some other Horn Of African population (i.e., the Ethio–Somali ancestry in the Ari Cultivators is likely to substantially postdate the initial entry of this ancestry into the region)."

van Dorp et al.[16] found similar ADMIXTURE results. Interpreted according to the protocol above, these analyses all imply that the Blacksmiths are pure representatives of one ancestral population (as shown by a homogeneous block of colour), while Cultivators are recently admixed, receiving ancestral contributions from neighbouring Ethiopian groups. However, the results of the three studies have different sampling and differ in how much of the ancestral population that Blacksmiths purportedly represent has contributed to the Cultivators or to other groups.

van Dorp et al.[16] used additional analyses including Ghost and Recent Bottleneck simulations, as in Fig. 2, together with fineSTRUCTURE[11], and GLOBETROTTER[12] to show that this history is false and the totality of evidence from the genetic data supports that the true history is analogous to the Recent Bottleneck scenario. The Blacksmiths and the Cultivators diverged from each other, principally by a bottleneck in the Blacksmiths, which was likely a consequence of their marginalised status. Once this drift is accounted for the Blacksmiths and Cultivators have almost identical inferred ancestry profiles and admixture histories. In our analysis, a strong deviation from a simple admixture model can be seen in the residual palettes, which imply that the ancestral palettes estimated by badMIXTURE substantially underestimate drift in the Ari Blacksmiths (Fig. 4e).

**Case Study 3: Worldwide human data**. An important consideration in any STRUCTURE analysis is sample size. This is vividly illustrated by the analyses of Friedlaender et al.[18] who augmented a pre-existing microsatellite data set from a worldwide collection by a similar number of samples from Melanesia, in

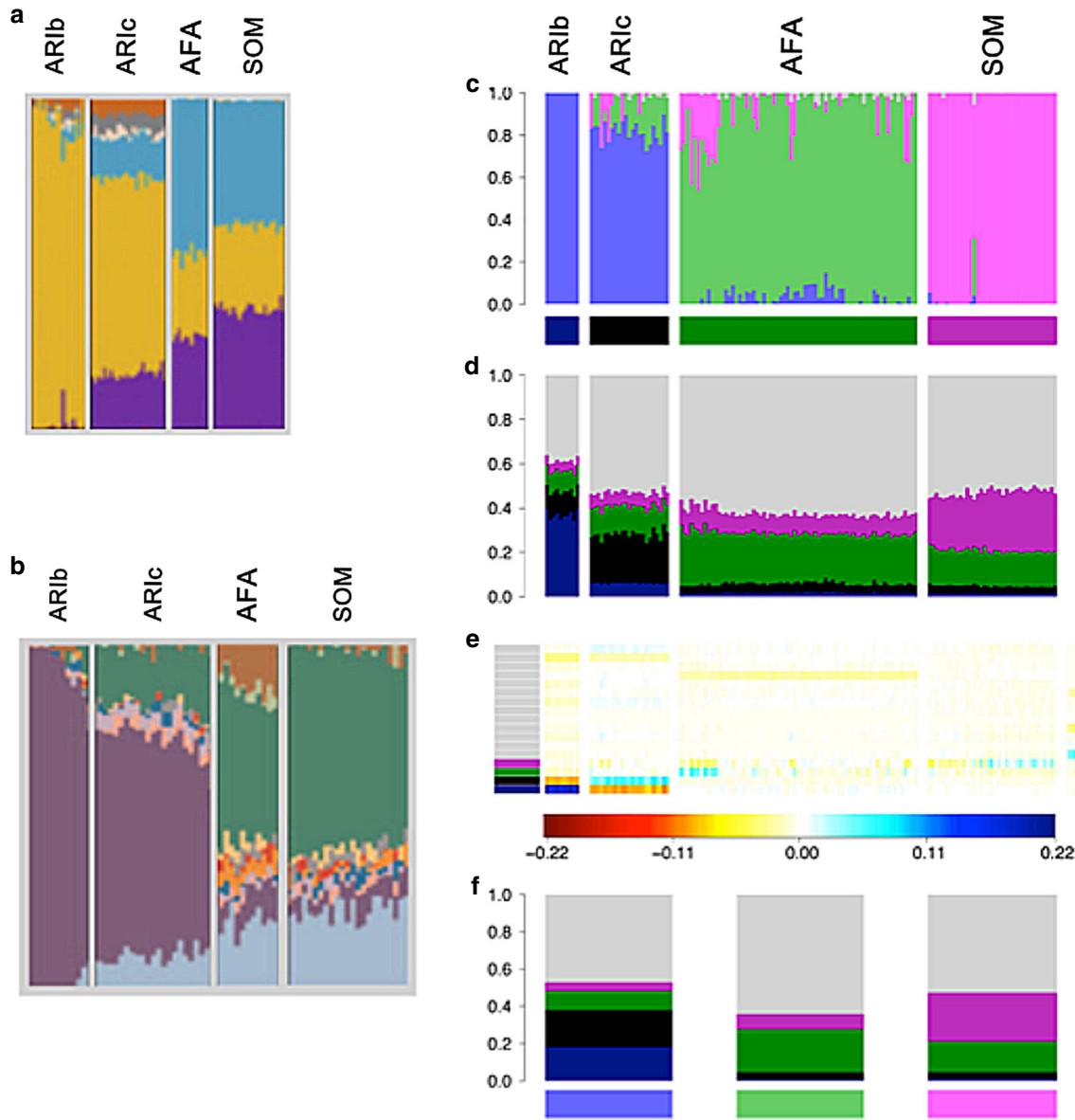

**Fig. 4** Analysis of Ari ancestry. ADMIXTURE analyses of the Ari and neighbouring Ethiopian groups adapted from **a** Pagani et al.[15], **b** Hodgson et al.[17], and **c** van Dorp et al.[16] at $K = 11$. Somali (SOM) and Afar (AFAR), Ari Blacksmith (ARIb) and Ari Cultivator (ARIc) populations were used in all three of the studies but the other populations differ substantially and the exact individuals differ slightly due to different quality control procedures and data set merges. **d** CHROMOPAINTER inferred painting palettes based on **c**. **e** badMIXTURE palette residuals under the best fit ancestral population admixture model. **f** Estimated ancestral palettes. Contributions from other populations are shown in grey

order to study genetic relationships between Melanesians, for which purpose their sample was excellent. For $K = 2$, their analysis infers Papua New Guinea (PNG) as one ancestral population and Western Eurasia and Africa as the other, with East Asians being represented as genetic mixtures (Fig. 5b). This analysis differs from that of Rosenberg et al.[19] for $K = 2$ who had only a small number of Melanesians in their sample, and who found Native Americans rather than Melanesians to be the unadmixed group (Fig. 5f). For $K = 6$, both models distinguish between all 5 continental groups (Americans, Western Eurasians, Africans, East Asians, and Oceanians). Rosenberg et al. split Native American groups into two ancestral populations while Friedlaender et al. infer that Melanesians have two ancestral populations (Fig. 5a). Rosenberg et al.[4] also found the Kalash, an isolated population in Pakistan, to be the sixth cluster.

For $K = 2$ both sets of results, interpreted literally, imply a continuous admixture cline. From almost any perspective, the most important demographic event that has left a signature in the data set is the out-of-Africa bottleneck. This is not taken by STRUCTURE to be the event at $K = 2$ in either of the analyses, or that of others with similar data sets, because sub-Saharan Africans constitute only a small proportion of the sample.

Some even more peculiar results are obtained for an analysis that focused on Melanesian populations, leaving in only East Asian populations and a single European population, the French. Friedlander et al.'s purpose in presenting this analysis was to analyse the fine-scale relationships amongst the Melanesians whilst accounting for admixture. Our purpose here is to ask what the results imply, when interpreted literally, about the relationships between Melanesians, East Asians and Europeans. For all

values from $K = 2$ to $K = 9$, the French population is inferred to be a mixture between an East Asian population and a Melanesian one (Fig. 5d, e). Only for $K = 10$ do the French form their own cluster and still have variable levels of admixture from East Asians (Fig. 5c). Throughout, interpretation of the ancestral populations based on where individuals are geographically today

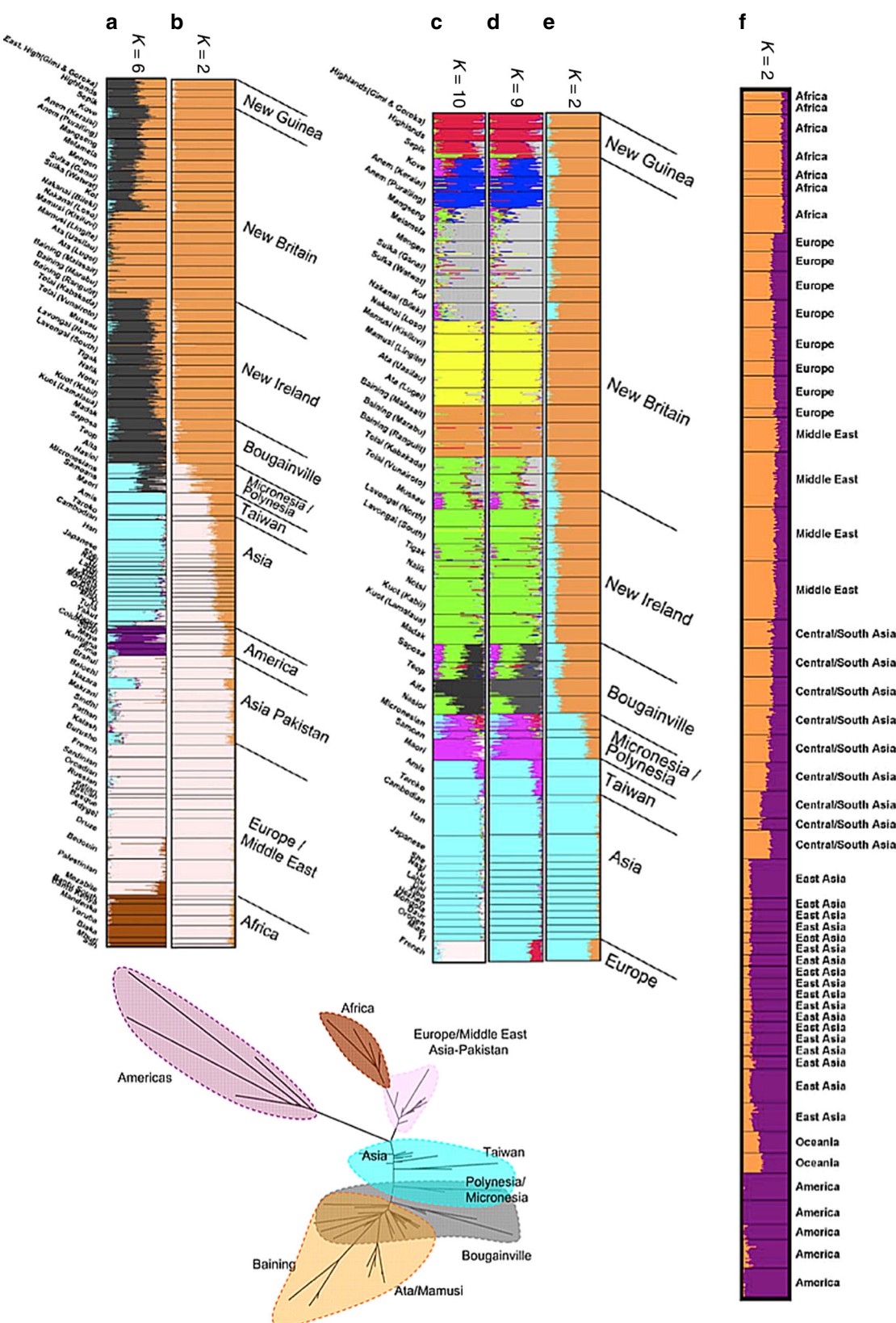

**Fig. 5** STRUCTURE results for global human genetic data sets. Panels **a–e** reproduced from Friedlaender et al.[18] and **f** from Rosenberg et al.[19]. **g** reproduces the neighbour-joining $F_{st}$ tree[18] coloured according to $K = 6$ STRUCTURE results in **a**

(Interpretation Protocol of Fig. 1) would only make these results more misleading, implying at $K = 9$ that the French are admixed between East Asians and Papuan highlanders.

It is tempting to write these results off as being the product of the sampling scheme, but the problem is fundamental to any approach based on equally weighing samples. If we instead imagine that there was an environmental catastrophe that spared the people of Melanesia and a few lucky others, then the analysis would become a faithful sampling of the people of the world. The results would become the world's genetic history but the literal interpretation of the bar plots would clearly be misleading, despite using sampling proportionate to extant humans.

The problem of sampling strategy affecting inference is common to many methods. Principle Components Analysis (PCA) is closely related to the STRUCTURE model in the information that it uses, both in theory[11] and in practice[20] and has also been shown theoretically to be affected by sample size[8]. Similarly, the neighbour-joining tree based on $F_{st}$ between populations from Friedlaender et al. also exaggerates the effect of drift, reproduced in Fig. 5.

The exercise highlighted by this case study is relevant in particular because human history is in fact full of episodes in which groups, such as the Bantu in Africa, the Han in Asia, and the Northern Europeans in America have used technological, cultural or military advantage or virgin territory to multiply until they make up a substantial fraction of the world's population. The history of the world told by STRUCTURE or ADMIXTURE is thus a tale that is skewed towards populations that are currently large and that have grown from small numbers of founders, with the bottlenecks that that implies.

**Case study 4: Ancient Indian populations**. In our final example, we attempt to address the challenge of complex inference by providing an overview of a demographic history in a single figure.

Basu et al.[21] used an ADMIXTURE plot with $K = 4$ to summarize variation amongst continental Indians from 19 labelled groups. The four ancestral populations were labelled Ancestral North Indian (ANI), Ancestral South Indian (ASI), Ancestral Tibeto-Burman (ATB) and Ancestral Austro-Asiatic (AAA), as shown in Fig. 6a. They argued that a major conclusion from their analysis is that the structure of mainland India is best described by 4 ancestral components.

The overall fit of the ADMIXTURE results estimated by badMIXTURE is poor (Fig. 6b). However, the large residuals are primarily within ancestral components, i.e., structured in a block-diagonal form, with blocks that correspond to the four ancestral components estimated by ADMIXTURE. Furthermore, almost all of the positive residuals are on the diagonal, i.e., specific to the labelled group that the individual was assigned to. These residuals vary substantially according to ancestral component, with ASI populations having the highest on-diagonal residuals and the ANI populations having the lowest ones. Within the KSH (Kharti) population, there is a substantial variation amongst individuals, presumably reflecting the presence of relatives or other strong sub-structure within this labelled group.

The structure of these residuals suggests that they principally reflect recent genetic drift that is specific to labelled groups, with considerable variation amongst groups in how much drift has occurred, presumably reflecting their recent demographic history. However, the block-like form suggests that if this recent genetic drift can be accounted for, the data might still be consistent with a history of mixture of four ancestral components, as suggested by the initial ADMIXTURE results.

We have implemented a simple procedure within badMIX-TURE to estimate the composition of painting palettes in the absence of group specific drift (see "Methods"), which for this data set substantially reduces the residuals (not shown). Fig. 6c shows the corrected individual palettes, and the ancestral palettes are substantially altered by the removal of recent drift, particularly for ASI and AAA populations (compare Fig. 6d, e). A more rigorous but laborious approach to removing label-specific drift, namely to remove individuals with the same label from the donor panel used for chromosome painting, was implemented by van Dorp et al.[16].

Examining the corrected palettes shown in Fig. 6c carefully, it is possible to see evidence that there are indeed four distinct ancestral components in the data, validating the major claim made by Basu et al. in their original analysis. For example, the three labelled groups with high ASI ancestry have similar palettes that are clearly distinct from those of all other labelled groups, with all of them having large amounts of green of three different shades. However, comparison of these palettes with the ADMIXTURE results also highlights the likely effect of recent genetic drift on those results, which is analogous to, but less dramatic than, that observed in the Ari case study. Specifically, the PNY (Paniya) are inferred by ADMIXTURE to be the only unadmixed representatives of the ASI population (Fig. 6a) but also have the largest badMIXTURE residuals prior to correction, which presumably reflects recent drift (Fig. 6b). After correction, PNY actually receive a smaller proportion of their palette from ASI groups than the other two ASI groups do (Fig. 6c).

This analysis highlights the fact that the mixture fractions estimated by ADMIXTURE may be unreliable and that no individual group can be safely assumed to be pure representatives of the ancestral source. That said, the relative admixture fractions are more plausible for the other three ancestry components, since the labelled groups that are estimated as pure by ADMIXTURE, i.e., BIR (Birhor), KSH and TRI (Tripuri)/JAM (Jamatia) also have the highest contribution from ancestrally related groups within their painting palettes.

Careful examination of these palettes also provide evidence of sharing of ancestry between pairs of populations that is not predicted based on the four ancestral palettes (shown above the black line in Fig. 6c), providing further evidence of the importance of recent demography, rather than ancestral population mixture, in shaping diversity. These pairs of populations are TRI and JAM, IRL (Irula) and KDR (Kadar), HO (Ho) and SAN (Santal) and BIR and KOR (Korwa). This sharing is most likely to have arisen during the divergence of the populations from each other. This might be due to shared drift or recent patterns of migration.

Overall, this analysis provides evidence for demographic events at multiple scales. At local and recent scales, there is evidence for heterogeneity within groups, as shown by individuals within labelled groups with atypical ADMIXTURE profiles (e.g., in TRI and JAM) or badMIXTURE residuals (e.g., in KSH). This heterogeneity provides evidence for recent migration between groups and substructure within groups, respectively. However, with the exception of some of the ANI populations, individuals in each group are distinguishable according to their badMIXTURE residuals and also based on fineSTRUCTURE clustering (see supplementary information of Basu et al.). This shows that most of the labelled groups are samples from populations that have been distinct from each other for long enough to acquire distinct and distinguishable genetic identities.

At the largest and most ancient scale, there is evidence for four ancestry components with clearly distinct painting palettes. However, the analysis in itself provides little evidence about the origin of these four ancestry components and the processes that gave rise to them, which would be best elucidated by relating the diversity found in India to that found in a global reference panel, together with demographic modelling.

The greatest challenges in model interpretation occur at intermediate scales. There is clear evidence for admixture between

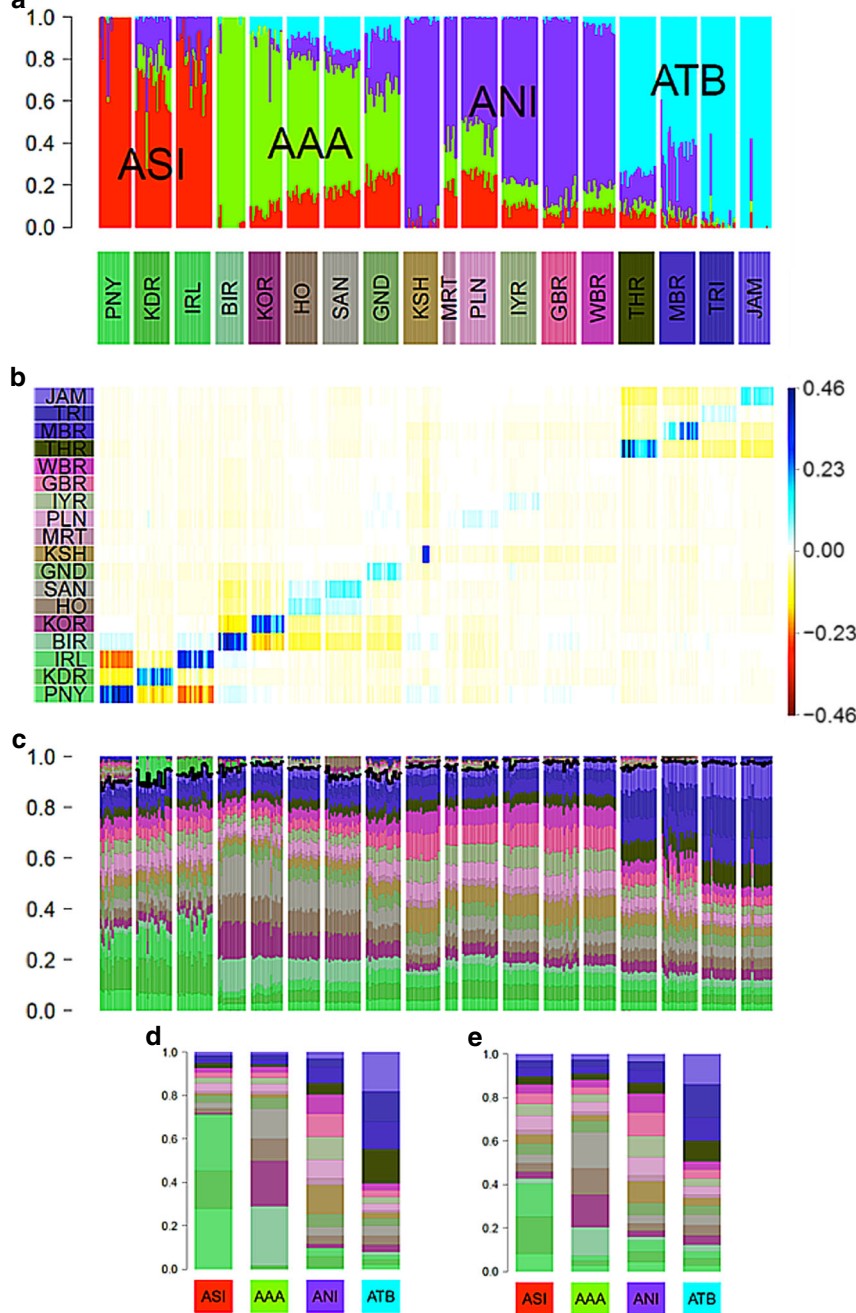

**Fig. 6** Comparison of ADMIXTURE with painting palettes for Indian genetic data originally presented in ref. [21]. **a** ADMIXTURE profile at $K = 4$. **b** Residuals palettes estimated by badMIXTURE. **c** Painting palettes after correcting within-population values as described in text. The part of the palette above the black line is not predicted by badMIXTURE. **d** Ancestral palettes estimated by badMIXTURE. **e** Estimated ancestral palettes after correcting for within-population values

ancestry components in some of the populations, such as GND (Gond) and MBR (Manipuri Brahmin). However, interpretation is made harder by the effects of recent genetic drift on ADMIXTURE estimates, seen most clearly for ASI populations. A much more involved analysis would be necessary to elucidate these migration patterns and to relate them to the overall population structure.

## Discussion

STRUCTURE and ADMIXTURE are popular because they give the user a broad-brush view of variation in genetic data, while allowing the possibility of zooming down on details about specific individuals or labelled groups. Unfortunately it is rarely the case that sampled data follows a simple history comprising a differentiation phase followed by a mixture phase, as assumed in an ADMIXTURE model and highlighted by case study 1. Naïve inferences based on this model (the Protocol of Fig. 1) can be misleading if sampling strategy or the inferred value of the number of populations $K$ is inappropriate, or if recent bottlenecks or unobserved ancient structure appear in the data. It is therefore useful when interpreting the results obtained from real data to think of STRUCTURE and ADMIXTURE as algorithms that parsimoniously explain variation between individuals rather than as parametric models of divergence and admixture.

For example, if admixture events or genetic drift affect all members of the sample equally, then there is no variation between individuals for the model to explain. Non-African humans have a few percent Neanderthal ancestry, but this is invisible to STRUCTURE or ADMIXTURE since it does not result in differences in ancestry profiles between individuals. The same reasoning helps to explain why for most data sets—even in species such as humans where mixing is commonplace—each of the $K$ populations is inferred by STRUCTURE/ADMIXTURE to have non-admixed representatives in the sample. If every individual in a group is in fact admixed, then (with some exceptions[14]) the model simply shifts the allele frequencies of the inferred ancestral population to reflect the fraction of admixture that is shared by all individuals.

Several methods have been developed to estimate $K$[1,2,22], but for real data, the assumption that there is a true value is always incorrect; the question rather being whether the model is a good enough approximation to be practically useful. First, there may be close relatives in the sample which violates model assumptions[23]. Second, there might be "isolation by distance", meaning that there are no discrete populations at all[24]. Third, population structure may be hierarchical, with subtle subdivisions nested within diverged groups. This kind of structure can be hard for the algorithms to detect and can lead to underestimation of $K$[25]. Fourth, population structure may be fluid between historical epochs, with multiple events and structures leaving signals in the data[12]. Many users examine the results of multiple $K$ simultaneously but this makes interpretation more complex, especially because it makes it easier for users to find support for preconceptions about the data somewhere in the results.

In practice, the best that can be expected is that the algorithms choose the smallest number of ancestral populations that can explain the most salient variation in the data. Unless the demographic history of the sample is particularly simple, the value of $K$ inferred according to any statistically sensible criterion is likely to be smaller than the number of distinct drift events that have practically impacted the sample. The algorithm uses variation in admixture proportions between individuals to approximately mimic the effect of more than $K$ distinct drift events without estimating ancestral populations corresponding to each one. In other words, an admixture model is almost always "wrong" (Assumption 2 of the Core protocol, Fig. 1) and should not be interpreted without examining whether this lack of fit matters for a given question.

Because STRUCTURE/ADMIXTURE accounts for the most salient variation, results are greatly affected by sample size[26] in common with other methods[8,27]. Specifically, groups that contain fewer samples or have undergone little population-specific drift of their own are likely to be fit as mixes of multiple drifted groups, rather than assigned to their own ancestral population. Indeed, if an ancient sample is put into a data set of modern individuals, the ancient sample is typically represented as an admixture of the modern populations (e.g., ref. [28,29]), which can happen even if the individual sample is older than the split date of the modern populations and thus cannot be admixed.

Here, as well as highlighting the problems, we have introduced a new tool, badMIXTURE which can be used to assess the model fit of STRUCTURE/ADMIXTURE results for each individual. The popularity of STRUCTURE and its descendants as unsupervised clustering methods means that they will be applied and interpreted, for which badMIXTURE provides important assistance. However, these analyses should always be followed up with tests of specific hypotheses, using other approaches. Running STRUCTURE or ADMIXTURE is the beginning of a detailed demographic and historical analysis, not the end.

## Methods

**Simulations.** Figure 2a illustrates the demographic histories behind three simulation scenarios we name "Recent Admixture", "Ghost Admixture" and "Recent Bottleneck".

These simulations comprise some of a subset of full simulations described in ref.[16], which aim to capture global human population genetic diversity across 13 simulated world-wide populations. Here, for tractability and motivated by Case Study 2, we explore the impact of different demographic histories in a subset of simulated groups: P1-P4. The simulation protocol used to generate the world-wide out-group populations is described in full detail in Supplementary Note 1.

For the "Recent Bottleneck" and "Ghost Admixture" simulations 10 populations were simulated using the approximate coalescence simulation software MaCS[30] under histories that differ in how P2 relates to P1 (Fig. 2a). For "Recent Bottleneck", P1 splits from P2 20 generations ago followed immediately by a strong bottleneck in P2. In "Ghost Admixture", instead P1 splits from P2 1700 generations ago, after which migrants from P1 form ~50% of P2 over a period of 200–300 generations. Although simulating 100 individuals in each population, we perform subsequent ADMIXTURE and CHROMOPAINTER analyses on a subset of these using only 35 individuals from P1, 25 individuals from P2, 70 individuals from P3 and 25 individuals from P4. This leaves an 'excess' of simulated individuals. For ease of interpretation only P1-P4 are depicted in Fig. 2 with all out-group populations coloured grey.

For the "Recent Admixture" scenario, we implement a simulation technique adapted from that applied in[31], related to that in[32], which sub-samples chromosomes from the 'excess' individuals simulated under the "Recent Bottleneck" scenario. This method explicitly mixes chromosomes from different populations based on a set of user-defined proportions, analogous to an instantaneous admixture event. Importantly for our purposes, this allows direct assessment of how well ADMIXTURE recapitulates these proportions. Using this approach, we simulate admixed chromosomes of P2 by mixing chromosomes of 20 'excess' individuals from each of P1 (50%), P3 (35%) and P4 (15%) based on an admixture event occurring $\lambda = 15$ generations ago. In particular, to simulate a haploid admixed chromosome and as in Leslie et al[31], we first sample a genetic distance $x$ from an exponential distribution with rate 0.15 ($\lambda/100$). The first $x$ cM of the simulated chromosome is composed of the first $x$ cM of chromosomes selected randomly, but without overlap, from 'excess' individuals of P1, P3, and P4 according to the defined proportions. This process is repeated using a new genetic distance sampled from the same exponential distribution (rate = 0.15) and continued until an entire simulated chromosome is generated. The method is then re-employed to generate a set of 20 haploid chromosomes for a single individual and then repeated 70 times to generate 70 haploid autosomes. Diploid individuals are constructed by joining two full sets of haploid chromosomes, resulting in 35 simulated P2 individuals in total.

**Estimation of ADMIXTURE bar plots and CHROMOPAINTER palettes.** For each simulation scenario we apply ADMIXTURE[2] to the sampled individuals from every simulated group. SNPs were first pruned to remove those in high linkage disequilibrium (LD) using PLINK v1.07[33] so that no two SNPs within 250 kb have a squared correlation coefficient ($r^2$) greater than 0.1. ADMIXTURE was then run with default values for multiple values of $K$, and the resultant admixture profiles plotted, where $K = 11$ (Figs. 2b and 4c). In addition, for each scenario, we applied CHROMOPAINTER to paint all individuals in relation to all others using default values for the CHROMOPAINTER mutation/emission ("-M") and switch ("-n") rates. When running CHROMOPAINTER ignoring information from Linkage Disequilibrium we use the unlinked mode ("-u"). We sum the total proportion of genome-wide DNA (linked) or matching chunk counts (unlinked) each recipient individual is painted by each donor group and plot the inferred contributions for each recipient as a painting palette.

**Estimation of ancestral palettes.** Define $A$ as the $N \times k$ admixture proportion matrix, where there are $N$ individuals in the sample and $K$ ancestral populations used in the ADMIXTURE analysis. Let $C$ be the $N \times P$ matrix of individual palettes from the CHROMOPAINTER painting, and $X$ be the $K \times P$ matrix of the palettes for each ancestral population. Then we seek solutions for $X$ that minimise the squared prediction error of the form:

$$AX = C.$$

We define $B = (A^T A)^{-1} A^T$. Then, $BAX = (A^T A)^{-1} A^T A X = X$, leading to the solution

$$X = (A^T A)^{-1} A^T C.$$

Note that there is no guarantee that $X$ will be positive. Negative elements would imply a poor fit of the admixture model, and alternative minimisation strategies might be employed to find $X$ subject to the constraint. Further, if the matrix $A^T A$ is rank deficient its inverse will not exist. This should only be the case if $K$ is chosen too large, or there are genuine symmetries in the data.

For a recent admixture model, long haplotypes are inherited from each of the donating populations in a given admixture proportion. If we assume that ancestral boundaries can be inferred then, excluding drift in either SNP frequency or haplotype structure, the palettes of admixed individuals are (by definition) a mixture with the same ancestry proportions as the SNPS under which admixture is inferred.

In this article, we use sampling labels to identify groups with distinct ancestry profiles, but if these are not available or are not predictive of genetic relationships, it is possible to use fineSTRUCTURE[11] to cluster individuals into genetically homogeneous groups based on their inferred palettes, thus generating labels.

**Data availability**. All materials used in this paper are available at github.com/danjlawson/badMIXTURE. The R code badMIXTURE is licenced under a GPLv3 licence. mixPainter, which performs the chromosome painting, is free for academic use only. The simulated data are also provided under GPLv3.

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

## Acknowledgements

This paper was stimulated by a discussion with Chuck Langley and also by the 2016 Workshop for Population and Speciation Genomics, Cesky Krumlov. We thank Jonathan Pritchard for suggesting the residuals plot and Analabha Basu, Kimberly Gilbert, Matthew Hahn, Razib Khan, Partha Majumder, Iain Mathieson, and Matthew Stephens for comments. D.F. is funded by a Medical Research Council Fellowship as part of the MRC CLIMB consortium for microbial bioinformatics (grant number MR/M501608/1). L.v.D. is supported by the Newton Trust (MR/ P007597/1). D.J.L. is supported by Wellcome Trust and Royal Society grant WT104125AIA.

## Author contributions

D.F. conceived the study. D.J.L. implemented badMIXTURE. L.v.D. performed simulations. All authors contributed to experimental design and to all stages of manuscript writing.

## Additional information

**Competing interests:** D.J.L. is a director of GENSCI Ltd. The remaining authors declare no competing interests.

