## [Peer Review File · Nature Communications]

Reviewers' comments:

Reviewer #1 (Remarks to the Author):

The current paper by Lawson, van Dorp and Falush represents an attempt to highlight some pitfalls when applying population genetic clustering algorithms and software (e.g. ADMIXTURE) for genomic data, and proposes a new approach to assess the clustering results.

The new metric for assessing clustering seems interesting and potentially useful, but unfortunately the paper as presently written is meandering, occasionally ironic in tone about guidelines, and occasionally slightly vitriolic in its description of the previous literature. In my view it also misrepresents the present research paradigm, sophisticated researchers do not follow the supposed step-by-step procedure the authors (correctly) outline as problematic, though some misguided studies may.

If the paper was substantially rewritten and focused to assess the new metric based on chromopainter in a systematic way, it could still reiterate its (valid) points about clustering analyses for the non-expert reader. However, a still unsolved question in such a refocused manuscript would be whether the proposed badMIXTURE method would be adopted for exactly the same purpose that the authors seemingly object to in the beginning of the paper: being used to estimate or motivate a choice of the "true K", without being on firm statistical footing in terms of model testing.

A more firm addressing of the problems with hierarchical models for the non-hierarchical modelling in clustering methods that the authors highlight in Figure 1 would be actually infer a hierarchical model such as the class of models simulated. In some sense, it could not be expected for clustering methods, neither ADMIXTURE, Structure, or fineSTRUCTURE, to confidently address such simulated models that have strong qualitative deviations from their underlying algorithms. Admixture graphs such as Treemix, qpgraph, and MixMapper do model such histories, and could certainly use an expanded literature on the statistical soundness of their model-fitting and model assessment, a research direction which would possibly be more fruitful to address the issues in the literature that the authors aim to address here.

Specific comments.

The paper's Figure 1 is a good visualization of how clustering results should not be taken at face value, for example since they model a starlike phylogeny where all modelled populations diverge instantaneously from each other. Clustering methods rely on a non-hierarchical models and thus have difficulty distinguishing the shared genetic drift in a hierarchical model where populations diverge at different times, from shared genetic drift arising from admixture. This is useful and bears repeating, but is a point that is known to most sophisticated population geneticists, and have been highlighted in the literature before (e.g. in Pickrell and Reich 2013, and Pickrell et al. 2014 supplementary information). This lack of modelling of hierarchical relationships is a main reason for why admixture graphs are now frequently used to model population history, while clustering methods are seen more as a non-parametric exploratory tool such as PCA, or for inferring admixture proportions in recent historical times where there is little drift since mixture.

The approach to use CHROMOPAINTER to assess clustering results is certainly interesting. However, the recent admixture scenario in Figure 1 is truly an idealized scenario where absolutely no drift has occurred since admixture. I am thus wondering whether in practise there will be clear separation in badMIXTURE between cases of poor clustering fit and cases that fit the model. It seems that more simulations finding the "breaking point" for where such a fine line exists may be necessary, as well as more empirical analyses showing how the authors envision that it should be applied.

Reviewer #2 (Remarks to the Author):

In this manuscript, Lawson and colleagues present results indicating that the simple model implemented in STRUCTURE and ADMIXTURE may fail to recognize admixture events. They further propose that in certain cases the use of methods that exploit haplotype information such as CHROMOPAINTER are better suited to identify admixture events (or the lack thereof). Finally, they propose to compare results obtained with CHROMOPAINTER and STRUCTURE/ADMIXTURE to assess the goodness of fit of structure models.

Given the wide usage of STRUCTURE/ADMIXTURE, a careful assessment of their interpretability is important. But while certainly interesting, this manuscript is a little unconventional. Specific general issues I have include:

1) The key finding from the simulation results are already published pretty much as is in van Dorp et al. (2015), including the interpretation regarding the Ethiopian data. The only aspect that is new is the use of their goodness of fit assessment. It appears to be strange to essentially write a second paper on pretty much the same data with the same message, yet it is true that many users of STRUCTURE / ADMIXTURE are unlikely to come across the van Dorp et al. (2015) paper. Also, it remains unclear how relevant the proposed goodness of fit test will be since it requires a linkage map. As the authors show, without such a map, CHROMOPAINTER fails to provide complementary information.

32 The manuscript portrays a narrow view on the use of structure models. While the summarized protocol given has surely been used a lot in the literature, there is also a huge number of researchers that used STRUCTURE / ADMIXTURE very differently, for instance by showing results for multiple K or by not interpreting all mixtures as admixture events. Indeed, these methods are also used a lot to identify genetic similarities or individuals that have an ancestry profile different from others at the same sampling location. The authors need to change their tone and to do justice to the plethora of users that applied these methods correctly, i.e., that saw these as clustering algorithms, not methods to prove / disprove admixture.

Similarly, there are plenty of papers that are very much interested in addressing some or all of the supplementary assumptions listed at the end of this section. It is fair to say that many users proceeded as mentioned, but you must do justice to people that investigated some or several of these additional assumptions, for instance by inferring the relationships using coalescent modeling or additional tools such as TreeMix, which is commonly done. Also, there is specific software to address some of your assumptions, such as for instance TreeMix, F-statistics or Ohana that infers specifically 3d) along with structure bars. It would be good to extend your discussion to mention the use of such tools to further investigate populations as admixed.

Also, I fail to see how the quote from Evanno et al. implies that these authors assumed that clusters are biologically meaningful in all situations. The out-of-context quote is taken from a paper that uses simulations to assess the power of STRUCTURE in identifying clusters, and "true" here simply refers to the number of demes simulated. Evanno et al. caution very much that their results can not be generalized to other settings.

Finally, many authors have conducted simulation studies to assess the power of STRUCTURE / ADMIXTURE and reported underestimation of K (including Evanno et al.), and more importantly how isolation-by-distance breaks the model. Please appropriately refer to previous studies illustrating the limits of these analyses.

5) Since the simulations are fundamental to all analyses and conclusions presented, I must insist

that the simulation procedure must be explained in the methods in sufficient detail to understand all important aspects without referring to van Dorp et al. (2015). That is particularly true regarding the idea of simulating 13 populations while only using four, but also regarding the actual simulation parameters such as all divergence times and population sizes.

Regarding the analysis of 13 populations: the authors make the claim that inferring admixture proportions with high numbers of K are challenging, even from many loci. I'm rather convinced that the issue presented hold under different scenarios as well, but since the authors made their case for using low K, to what degree are the results affected by their choice of simulating very many populations? This is relevant in the sense that the algorithm might be able to recognize P2 as an additional ancestry source if only these populations were studied (the authors hint at that by mentioning that more drift in P2 might lead to that finding). But importantly: all this is not relevant for the key message of the manuscript.

Minor comments

Line and page numbers would have been helpful. It is unfortunately now hard to refer to specific parts.

Abstract: "A successful example is THE reconstruction ..."

Page 3: If you want to use the term "assumptions" for your points 3a – 3d, then please phrase them as assumptions. "Do not ask ..." is not an assumption.

Page 5: "... often does is in practice is ..."

Bottom of Page 7: you probably refer to Figure 3 here.

Reviewer #3 (Remarks to the Author):

This paper provides some needed insights into the use and interpretation of STRUCTURE and ADMIXTURE analyses. I see the authors making three critical points:

1. The population assignments that result from these analyses are best thought of in terms of "drift events" rather than admixtures (*sensu stricto*).
2. The extent to which true admixture is a sufficient hypothesis can be investigated through the use of reconstructed palettes.
3. Sample configuration can drive clustering.

I find the content of the paper both useful and compelling. I'm making the recommendations below as suggestions to increase the readability and usability of this paper.

pg. 4, par. 2 ("In practice...") This paragraph could stand to be expanded for better clarity, and there are a few words and phrases in it that may be misleadingly specific. "Most salient variation" and "have significantly impacted" both stick out in this regard. Fleshing out exactly what is meant here would be more helpful.

pg 4, par 3. The assertion of the last sentence (starting "The algorithm is more likely...") strikes me as something that may depend on the sample size. Is this the case?

pg 5, par 3. When you say "true palettes," is this truth from simulation or inferred from data?

The last sentence needs more explanation. It is not clear exactly what "this" refers to, and how we are to know that it is an indication.

pg 5 par 4. To what extent does the argument in this paragraph hold when admixture proportions are small?

pg 6 par 3, 4, Mislabeled figure references.

pg 7 last par "based on several different analyses" I'd appreciate another sentence or two about what these were.

pg 8 par 1 "Sample size influences the results ... giving more weight to larger populations" Is this a desirable feature? Would the results be more stable if the residuals were re-weighted by sample size? Also, I think you mean larger samples, no larger populations.

pg 8 par 3 PNG has not been defined yet. Later you go back and forth between the abbreviation and the full name.

pg 9 par 3. ("Once again...") I wasn't sure how to interpret this paragraph. While the focus of this paper is on not taking STRUCTURE/ADMIXTURE results at face value, this paragraph seems to invite us to do just that. It is not clear if the outlined scenario is meant as a serious proposal or simply a thought exercise.

pg 11 par 2 The word "likely" occurs twice in here, but in both cases it is not clear what leads to these conclusions.

pg 11 par 3 While I understand that you do not trust the unadjusted version of figure 4c and therefore need to make this somewhat circuitous construct, I'd like to see the naive version and hear a bit more about why it should be dismissed. Should these kinds of adjustments be standard practice?

pg 11 par 4. I would like to see this paragraph expanded. I do not find fig 4c quite so readily legible and would appreciate better instruction about what features of it are being used to draw the inferences in this paragraph. I trust that there is "good evidence" that "can be seen by eye", but clear instruction about exactly what my eye should be looking at would help considerably.

Lastly, while I'm not certain this paper is the right vehicle for this, I would love to hear the authors' response to this tweet from Rasmus Nielsen
(https://twitter.com/ras_nielsen/status/714953486723473408)

"Gene-flow from [unsampled] 4th pop. into pop. 1 causes spurious evidence of 'admixture' in pop. 2. Simulation by @maltethodberg"

Inline image 1 (STRUCTURE plot with three pops, $k=2$, pop2 looking admixed)

Inline image 2 (Phylogeny (((pop1, pop2), pop3), pop4) with recent admixture from pop4 to pop1)

Reviewers' comments:

Reviewer #1 (Remarks to the Author):

The current paper by Lawson, van Dorp and Falush represents an attempt to highlight some pitfalls when applying population genetic clustering algorithms and software (e.g. ADMIXTURE) for genomic data, and proposes a new approach to assess the clustering results.

The new metric for assessing clustering seems interesting and potentially useful, but unfortunately the paper as presently written is meandering, occasionally ironic in tone about guidelines, and occasionally slightly vitriolic in its description of the previous literature.

Response: We have substantially reorganized the paper to present the ideas more cleanly.

(1) We have made a new figure 1, which contains the protocol, which is described as a caricature.

(2) We are now careful to avoid irony in the rest of the manuscript and vitriol anywhere. We explain that researchers tend to follow elements of the protocol because it is far from

obvious how else to interpret the results otherwise. We have acknowledged that the details we are picking up on here were often not the inference target in the original research.

(3) We have reorganized the manuscript to present the problems first and the solutions second.

(4) We have rewritten the badMIXTURE sections to focus more cleanly on how they help users to interpret structure/ADMIXTURE barplots.

In my view it also misrepresents the present research paradigm, sophisticated researchers do not follow the supposed step-by-step procedure the authors (correctly) outline as problematic, though some misguided studies may.

We agree that there are researchers who interpret the results in a sophisticated way. However sophisticated researchers frequently throw up their hands in despair at how STRUCTURE is commonly used. Our article is an attempt to increase the sophistication of the bulk of users, as is indicated by using the word "Tutorial" in the title.

We also think that interpreting statistical genetic analysis is really hard! Sophisticated users make mistakes as is vividly demonstrated

by the different analyses of the Ari that was performed by well-known human population geneticists who are amongst the most sophisticated users of these methods in the entire community. We believe we are now being fair to them as they were very careful in the way they presented interpretations of conclusions.

If the paper was substantially rewritten and focused to assess the new metric based on chromopainter in a systematic way, it could still reiterate its (valid) points about clustering analyses for the non-expert reader. However, a still unsolved question in such a refocused manuscript would be whether the proposed badMIXTURE method would be adopted for exactly the same purpose that the authors seemingly object to in the beginning of the paper: being used to estimate or motivate a choice of the "true K", without being on firm statistical footing in terms of model testing.

Response:

We have made numerous changes to focus the manuscript on a single purpose, namely to describe the practical challenges of interpreting a single model output for real data, where there are many unknown deviations from model assumptions. We highlight this through examples which are now presented a set of case

studies.

We disagree that this purpose could be achieved equally well in a paper focused on demonstrating the validity of a new metric, which would inevitably require the paper to be based around simulated data with known properties. However, we agree that the purpose of the previous version was somewhat confused and have addressed this as described in response to the previous point.

It would also be difficult to achieve the same purpose in a manuscript that discussed results from many different methods in depth at once. Every method has its pitfalls, and this would make the discussion far too complex. We have rewritten the badMIXTURE description to make it explicit that the purpose in the manuscript is to help users understand the STRUCTURE/ADMIXTURE barplots and their limitations.

We respect the reviewers point of view, and hope we have made a convincing case for our choice in the revised paper. We think that this comes down to an editorial decision about what subject matter is appropriate for Nature Communications. Appreciation for the distinctive and valuable contribution that our manuscript makes has been expressed to us in person and on twitter by many colleagues as well as in the published literature, for example by

Novembre Genetics 2016:

<http://www.genetics.org/content/204/2/391>

Another challenge is that *STRUCTURE* has become, in some sense, a victim of its own success. It is applied by default in most studies without consideration of whether the underlying model is relevant. For example, if applied to a geographic continuum, the method will infer source populations that are vaguely spatial but have no real interpretation as source populations in an admixed sample (e.g., Witherspoon et al. 2007). A recent paper captures the care needed with its colorful title: “A tutorial on how (not) to over-interpret STRUCTURE/ADMIXTURE bar plots” (Falush et al. 2016). All this being said, the need for careful interpretation is ubiquitous in population genetics, and the extra attention on the *STRUCTURE* method is warranted because of its widespread use.

Finally, we do not advocate users using badMIXTURE to infer “true K” and think this is unlikely to happen in practice.

A more firm addressing of the problems with hierarchical models for the non-hierarchical modelling in clustering methods that the authors highlight in Figure 1 would be actually infer a hierarchical model such as the class of models simulated. In some sense, it could not be expected for clustering methods, neither ADMIXTURE, Structure, or fineSTRUCTURE, to confidently address such simulated models that have strong qualitative deviations from their underlying algorithms. Admixture graphs such as Treemix, qpgraph, and MixMapper do model such histories, and could certainly use an expanded literature on the statistical soundness of

their model-fitting and model assessment, a research direction which would possibly be more fruitful to address the issues in the literature that the authors aim to address here.

Response: We have expanded our discussion of alternative methods, which we agree was far too abbreviated in the previous manuscript and continue to emphasize that running STRUCTURE/ADMIXTURE should be considered as being a starting point for analysis rather than an endpoint.

We also agree that there is plenty of interesting work to be done on development of new hierarchical and other models. However, once again, our manuscript is concerned with the practical problem of interpreting the model output that users have in front of them, which for very many users will just be STRUCTURE or ADMIXTURE rather than of determining the best possible tool for inference for any given problem. We think this would be an interesting avenue of future work but is beyond the scope of this particular paper.

Specific comments.

The paper's Figure 1 is a good visualization of how clustering results should not be taken at face value, for example since they

model a starlike phylogeny where all modelled populations diverge instantaneously from each other. Clustering methods rely on a non-hierarchical models and thus have difficulty distinguishing the shared genetic drift in a hierarchical model where populations diverge at different times, from shared genetic drift arising from admixture. This is useful and bears repeating, but is a point that is known to most sophisticated population geneticists, and have been highlighted in the literature before (e.g. in Pickrell and Reich 2013, and Pickrell et al. 2014 supplementary information). This lack of modelling of hierarchical relationships is a main reason for why admixture graphs are now frequently used to model population history, while clustering methods are seen more as a non-parametric exploratory tool such as PCA, or for inferring admixture proportions in recent historical times where there is little drift since mixture.

Response: we agree that admixture models are known to have problems to sophisticated readers but think our Figure (old:1, new:2) adds value for them in explicitly describing the non-identifiability, whilst also providing a model checking tool.

Admixture graph approaches are currently still only accessible to experts and don't fulfill the data visualization role of admixture, so we think that both will provide value. We also note that the

literature you mention, although making very valuable contributions, do not situate their narrative around a tool-kit to aid interpretation as we have aimed to do here and are exclusively discussing human data. In order to communicate to a wider body of users we believe a more direct and practical article is necessary.

The approach to use CHROMOPAINTER to assess clustering results is certainly interesting. However, the recent admixture scenario in Figure 1 is truly an idealized scenario where absolutely no drift has occurred since admixture. I am thus wondering whether in practise there will be clear separation in badMIXTURE between cases of poor clustering fit and cases that fit the model. It seems that more simulations finding the "breaking point" for where such a fine line exists may be necessary, as well as more empirical analyses showing how the authors envision that it should be applied.

Response: Our India example (now called case study 4) describes this, as we now try harder to explain. In short: there is not a clear separation, but badMIXTURE still allows a correct interpretation even in the presences of complex pre- and post-admixture substructure.

Reviewer #2 (Remarks to the Author):

In this manuscript, Lawson and colleagues present results indicating that the simple model implemented in STRUCTURE and ADMIXTURE may fail to recognize admixture events. They further propose that in certain cases the use of methods that exploit haplotype information such as CHROMOPAINTER are better suited to identify admixture events (or the lack thereof). Finally, they propose to compare results obtained with CHROMOPAINTER and STRUCTURE/ADMIXTURE to assess the goodness of fit of structure models.

Given the wide usage of STRUCTURE/ADMIXTURE, a careful assessment of their interpretability is important. But while certainly interesting, this manuscript is a little unconventional. Specific general issues I have include:

Response: We have attempted to address the "non-conventional" format to make the manuscript more structured and easier to read. See also our opening response to reviewer 1.

1) The key finding from the simulation results are already published pretty much as is in van Dorp et al. (2015), including the interpretation regarding the Ethiopian data. The only aspect that is

new is the use of their goodness of fit assessment. It appears to be strange to essentially write a second paper on pretty much the same data with the same message, yet it is true that many users of STRUCTURE / ADMIXTURE are unlikely to come across the van Dorp et al. (2015) paper. Also, it remains unclear how relevant the proposed goodness of fit test will be since it requires a linkage map. As the authors show, without such a map, CHROMOPAINTER fails to provide complementary information.

Response: the key difference is that van Dorp et al describes a single instance of a problem which we include as one case study, whereas this paper describes the general problem as it applies to other researchers and published case studies as well as offering a solution. Additionally we have performed additional simulations to those out-lined in van Dorp et al in order to illustrate an idealized recent-admixture example, and included a new piece of software.

We hope the revised version succeeds better in this goal but the added value is already clear to a large number of readers, since as an archive manuscript it has already overtaken van Dorp et al. in citations, as well as receiving 10 fold more downloads, with more than 7,700 to date, a very encouraging number.

We also note that the goodness of fit test does not require a linkage map (see old Fig2, new Fig3) and reiterate this important point in the text: "badMIXTURE can be applied to data that has no linkage map as well as exploiting linkage information to make a more sensitive measure."

32 The manuscript portrays a narrow view on the use of structure models. While the summarized protocol given has surely been used a lot in the literature, there is also a huge number of researchers that used STRUCTURE / ADMIXTURE very differently, for instance by showing results for multiple K or by not interpreting all mixtures as admixture events. Indeed, these methods are also used a lot to identify genetic similarities or individuals that have an ancestry profile different from others at the same sampling location. The authors need to change their tone and to do justice to the plethora of users that applied these methods correctly, i.e., that saw these as clustering algorithms, not methods to prove / disprove admixture.

Response: As noted in the response to reviewer 1, we agree broadly that the protocol is a caricature but that (because it is just taking the model at face value) this is frequently done. We agree that not everyone gets this wrong, and have tried to make this clear in the revised tone as well as stressing where these approaches work really well in our section entitled:

STRUCTURE/ADMIXTURE are excellent tools for analysing recent admixture between differentiated groups . *Also, we note that using multiple values of K can be valuable but that it can also potentially make interpretation problems worse, not better, by allowing the user to find results that fit better with his/her preconceptions.*

Similarly, there are plenty of papers that are very much interested in addressing some or all of the supplementary assumptions listed at the end of this section. It is fair to say that many users proceeded as mentioned, but you must do justice to people that investigated some or several of these additional assumptions, for instance by inferring the relationships using coalescent modeling or additional tools such as TreeMix, which is commonly done. Also, there is specific software to address some of your assumptions, such as for instance TreeMix, F-statistics or Ohana that infers specifically 3d) along with structure bars. It would be good to extend your discussion to mention the use of such tools to further investigate populations as admixed.

Response: We agree and have incorporated this point into the manuscript, as also noted in response to reviewer 1.

Also, I fail to see how the quote from Evanno et al. implies that these authors assumed that clusters are biologically meaningful in

all situations. The out-of-context quote is taken from a paper that uses simulations to assess the power of STRUCTURE in identifying clusters, and "true" here simply refers to the number of demes simulated. Evanno et al. caution very much that their results can not be generalized to other settings.

Response: We agree that we were unfair to Evanno et al. who were only providing a useful tool, that has not always been appropriately used. We have in any case removed this section as part of streamlining the manuscript.

Finally, many authors have conducted simulation studies to assess the power of STRUCTURE / ADMIXTURE and reported and underestimation of K (including Evanno et al.), and more importantly how isolation-by-distance breaks the model. Please appropriately refer to previous studies illustrating the limits of these analyses.

Our new paragraph on estimating K describes how isolation by distance, relatives and other factors break model assumption, giving the key reference for each one.

5) Since the simulations are fundamental to all analyses and

conclusions presented, I must insist that the simulation procedure must be explained in the methods in sufficient detail to understand all important aspects without referring to van Dorp et al. (2015). That is particularly true regarding the idea of simulating 13 populations while only using four, but also regarding the actual simulation parameters such as all divergence times and population sizes.

We have added a new Supplementary Note 1 which provides the full simulation parameters which are also illustrated by a supplementary figure.

Regarding the analysis of 13 populations: the authors make the claim that inferring admixture proportions with high numbers of K are challenging, even from many loci. I'm rather convinced that the issue presented hold under different scenarios as well, but since the authors made their case for using low K , to what degree are the results affected by their choice of simulating very many populations? This is relevant in the sense that the algorithm might be able to recognize P2 as an additional ancestry source if only these populations were studied (the authors hint at that by mentioning that more drift in P2 might lead to that finding). But importantly: all this is not relevant for the key message of the

manuscript.

Response: we agree. We've rephrased the discussion of inferring K to make the assumption violation clear: under-estimation of K is to be expected and has consequences, specifically making it less likely that the model can be taken at face value. (Figure 1, protocol 1)

Minor comments

Line and page numbers would have been helpful. It is unfortunately now hard to refer to specific parts.

Abstract: "A successful example is THE reconstruction ..."

Response: Corrected, thank you.

Page 3: If you want to use the term "assumptions" for your points 3a – 3d, then please phrase them as assumptions. "Do not ask ..." is not an assumption.

Response: Corrected, thank you.

Page 5: "... often does is in practice is ...

Response: Corrected.

Bottom of Page 7: you probably refer to Figure 3 here.

Response: Corrected.

Reviewer #3 (Remarks to the Author):

This paper provides some needed insights into the use and interpretation of STRUCTURE and ADMIXTURE analyses. I see the authors making three critical points:

1. The population assignments that result from these analyses are best thought of in terms of "drift events" rather than admixtures (sensu stricto).
2. The extent to which true admixture is a sufficient hypothesis can be investigated through the use of reconstructed palettes.
3. Sample configuration can drive clustering.

I find the content of the paper both useful and compelling. I'm making the recommendations below as suggestions to increase the readability and usability of this paper.

Response: Thank you for your positive comments!

pg. 4, par. 2 ("In practice...") This paragraph could stand to be expanded for better clarity, and there are a few words and phrases in it that may be misleadingly specific. "Most salient variation" and "have significantly impacted" both stick out in this regard. Fleshing out exactly what is meant here would be more helpful.

We now flesh out that we mean salient ancestry-related genetic variation, give examples that events that effect small proportions of the sample or have modest effects might have little effect and also relate this discussion to the specific part of the Protocol that is made problematic by these issues. The paragraph should now be much clearer.

pg 4, par 3. The assertion of the last sentence (starting "The algorithm is more likely...") strikes me as something that may depend on the sample size. Is this the case?

Response: Yes, this could be a sample size effect as we now explicitly mention.

pg 5, par 3. When you say "true palettes," is this truth from simulation or inferred from data?

Response: This should have read "painting palettes" rather than "true palettes". This section has been amended in the new version.

The last sentence needs more explanation. It is not clear exactly what "this" refers to, and how we are to know that it is an indication.

Response: Fixed, thank you.

pg 5 par 4. To what extent does the argument in this paragraph hold when admixture proportions are small?

We think it is still true but of course the issue of small admixture proportions interact with statistical power to detect them accurately, and less power again to detect model deviations associated with this. Therefore badmixture and our approach can provide less assistance interpreting small admixture proportions. Fortunately most researchers are usually appropriately cautious in this case.

pg 6 par 3, 4, Mislabeled figure references.

Response: fixed, thank you.

pg 7 last par "based on several different analyses" I'd appreciate another sentence or two about what these were.

Response: these are now detailed.

pg 8 par 1 "Sample size influences the results ... giving more weight to larger populations" Is this a desirable feature? Would the results be more stable if the residuals were re-weighted by sample size? Also, I think you mean larger samples, no larger populations.

Response: fixed.

pg 8 par 3 PNG has not been defined yet. Later you go back and forth between the abbreviation and the full name.

Response: fixed, thank you.

pg 9 par 3. ("Once again...") I wasn't sure how to interpret this paragraph. While the focus of this paper is on not taking

STRUCTURE/ADMIXTURE results at face value, this paragraph seems to invite us to do just that. It is not clear if the outlined scenario is meant as a serious proposal or simply a thought exercise.

Response: This is an attempt to see the logical consequences of a literal interpretation . It has been rewritten to make that clearer.

pg 11 par 2 The word "likely" occurs twice in here, but in both cases it is not clear what leads to these conclusions.

Response: This section has been substantially shortened and rewritten for clarity, hopefully making the conclusions more apparent.

pg 11 par 3 While I understand that you do not trust the unadjusted version of figure 4c and therefore need to make this somewhat circuitous construct, I'd like to see the naive version and hear a bit more about why it should be dismissed. Should these kinds of adjustments be standard practice?

Response: this is a good question. We've put the unadjusted version into a supplement and changed the discussion to address

this point. Adjustment certainly shouldn't be done in all cases – we now explain that it is appropriate when there are two timescales involved: a recent admixture (which is not the subject of analysis) and an ancient admixture which is. The drift removal allows assessment of the ancient admixture event.

pg 11 par 4. I would like to see this paragraph expanded. I do not find fig 4c quite so readily legible and would appreciate better instruction about what features of it are being used to draw the inferences in this paragraph. I trust that there is "good evidence" that "can be seen by eye", but clear instruction about exactly what my eye should be looking at would help considerably.

Response: This has now been rewritten to make the points much easier to read.

Lastly, while I'm not certain this paper is the right vehicle for this, I would love to hear the authors' response to this tweet from Rasmus Nielsen

(https://twitter.com/ras_nielsen/status/714953486723473408)

"Gene-flow from [unsampled] 4th pop. into pop. 1 causes spurious evidence of 'admixture' in pop. 2. Simulation by @maltethodberg"

Inline image 1 (STRUCTURE plot with three pops, $k=2$, pop2 looking admixed)

Inline image 2 (Phylogeny (((pop1, pop2), pop3), pop4) with recent admixture from pop4 to pop1)

Response: It is a great example isn't it! It is a shame that there isn't a clear pathway for unpublished work like this to interact with academic literature. We totally agree with the sentiment but don't see how to raise it except in our own use of social media.

REVIEWERS' COMMENTS:

Reviewer #1 (Remarks to the Author):

While the revised version of this manuscript shows increased coherence and decreased use of the irony that made the reading of the previous version confusing, I find that the central issues remain and make it unsuitable for publication in a widely read journal such as Nature Communications.

The points about misinterpretation are important for the wider community of biologists who use population genetics as a tool. The simulation experiments would e.g. be very illustrative in a review paper, since they in many ways should be basic knowledge for any empirical population geneticist. Perhaps in this sense this paper could be more suitable to be directed to an audience e.g. in ecology, but at the same time as referee 2 points out with regard to the previous version, the new badMIXTURE approach to validation requires the CHROMOPAINTER software and optimally a linkage map, something that is unlikely to be available yet for many non-model organisms.

The paper now recognizes that there is plenty of high-quality work that is fully aware of the shortcomings of non-hierarchical clustering approaches, but it remains unclear that an approach of clustering+badMIXTURE is preferable to the alternatives such as model-based inference using isolation-migration models (IM/MIMAR/ABC/fastsimcoal/DADI etc) or admixture graphs (Treemix/QPADM/MIXMAPPER) etc. In my view, clustering methods are a data exploration tool that sometimes allow limited hypothesis-testing with regards to ancestry. The model underlying these methods is unlikely to be useful for many population history inference purposes due to its simplistic assumptions of instantaneous divergence between all ancestral populations without subsequent gene flow. Thus it may be recommendable to dissuade the practice of making inferences about e.g. "the true K" instead of building tools that somewhat enhance the model-inference abilities of these approaches, but do not enhance them such that the inference outperforms hierarchical model inference with gene flow (no quantitative comparison to such inference methods have been added to the revised version).

To summarize, the authors exploration of the behaviour of clustering methods under different demographic histories will be useful reading especially for non-experts, but has also been explored before (e.g. in the not-cited Engelhardt & Stephens 2010 and other references cited by reviewer 2). The new badMIXTURE approach is potentially interesting but not clearly a significant advance in population genetic methods to understand population history. This paper's useful contributions, especially in the pedagogical simulation figures, would thus in my view be more suitable for a more specialized journal.

Reviewer #2 (Remarks to the Author):

I'm happy to see that the authors have taken the task to thoroughly revise their paper seriously. All my initial concerns have been appropriately addressed and I now agree that the manuscript will provide a helpful discussion on the use of structure/admixture analyses.

Despite the risk of sounding old fashioned, I can not refrain from noting that neither download statistics of a preprint nor the number of twitter comments are any meaningful measure of quality. Wasn't that the lesson of 2017?

Reviewer #3 (Remarks to the Author):

The revisions greatly clarify the paper and present the material in a more concise and readable fashion. I am satisfied with the adjustments.

REVIEWERS' COMMENTS:

Reviewer #1 (Remarks to the Author):

While the revised version of this manuscript shows increased coherence and decreased use of the irony that made the reading of the previous version confusing, I find that the central issues remain and make it unsuitable for publication in a widely read journal such as Nature Communications.

We are very glad that despite their overall doubts about the paper, the reviewer recognizes the improvements that we have made.

The points about misinterpretation are important for the wider community of biologists who use population genetics as a tool. The simulation experiments would e.g. be very illustrative in a review paper, since they in many ways should be basic knowledge for any empirical population geneticist. Perhaps in this sense this paper could be more suitable to be directed to an audience e.g. in ecology, but at the same time as referee 2 points out with regard to the previous version, the new badMIXTURE approach to validation requires the CHROMOPAINTER software and optimally a linkage map, something that is unlikely to be available yet for many non-model organisms.

We disagree because we think that human geneticists, model organism geneticists as well as ecologists often have trouble interpreting the bar plots.

The paper now recognizes that there is plenty of high-quality work that is fully aware of the shortcomings of non-hierarchical clustering approaches, but it remains unclear that an approach of clustering+badMIXTURE is preferable to the alternatives such as model-based inference using isolation-migration models (IM/MIMAR/ABC/fastsimcoal/DADI etc) or admixture graphs (Treemix/QPADM/MIXMAPPER) etc. In my view, clustering methods are a data exploration tool that sometimes allow limited hypothesis-testing with regards to ancestry. The model underlying these methods is unlikely to be useful for many population history inference purposes due to its simplistic assumptions of instantaneous divergence between all ancestral populations without subsequent gene flow. Thus it may be recommendable to dissuade the practice of making inferences about e.g. "the true K" instead of building tools that somewhat enhance the model-inference abilities of these approaches, but do not enhance them such that the inference outperforms hierarchical model inference with gene flow (no quantitative comparison to such inference methods have been added to the revised version).

We definitely do not make any kind of claim that clustering+badMIXTURE is preferable to other approaches at any point in the manuscript and indeed emphasize the need for other methods in both introduction and discussion. All methods have issues of interpretation and we continue to believe in to our approach of discussing the issues with one very popular type of method in depth represents a substantial contribution to the literature and are pleased that we have been able to convince the other two reviewers and editors of Nature Communications of this.

To summarize, the authors exploration of the behaviour of clustering methods under different demographic histories will be useful reading especially for non-experts, but has also been explored before (e.g. in the not-cited Engelhardt & Stephens 2010 and other references cited by reviewer 2). The new badMIXTURE approach is potentially interesting but not clearly a significant advance in population genetic methods to understand population history. This paper's useful contributions, especially in the pedagogical simulation figures, would thus in my view be more suitable for a more specialized journal.

We agree that not citing Englehardt and Stephens was an omission, which we have corrected.

Reviewer #2 (Remarks to the Author):

I'm happy to see that the authors have taken the task to thoroughly revise their paper seriously. All my initial concerns have been appropriately addressed and I now agree that the manuscript will provide a helpful discussion on the use of structure/admixture analyses.

Despite the risk of sounding old fashioned, I can not refrain from noting that neither download statistics of a preprint nor the number of twitter comments are any meaningful measure of quality. Wasn't that the lesson of 2017?

We completely agree of course that download statistics and the number of twitter comments are not themselves indicators of quality, although they might correlate loosely with modern buzzwords "timeliness" "impact" etc, and possibly even relevance, which is closer to the argument we were trying to advance. More seriously, we have been very encouraged by the substance of the comments that people have made about the manuscript, on twitter and elsewhere, such as "I wish I read this paper before doing my phd."

Reviewer #3 (Remarks to the Author):

The revisions greatly clarify the paper and present the material in a more concise and readable fashion. I am satisfied with the adjustments.

We are glad that all three reviewers recognized the efforts we went to in revising our manuscript.